# Sample Complexity of Learning Mahalanobis Distance Metrics

**Nakul Verma**
Janelia Research Campus, HHMI
`verman@janelia.hhmi.org`

**Kristin Branson**
Janelia Research Campus, HHMI
`bransonk@janelia.hhmi.org`

## Abstract

Metric learning seeks a transformation of the feature space that enhances prediction quality for a given task. In this work we provide PAC-style sample complexity rates for supervised metric learning. We give matching lower- and upper-bounds showing that sample complexity scales with the representation dimension when no assumptions are made about the underlying data distribution. In addition, by leveraging the structure of the data distribution, we provide rates *fine-tuned* to a specific notion of the intrinsic complexity of a given dataset, allowing us to relax the dependence on representation dimension. We show both theoretically and empirically that augmenting the metric learning optimization criterion with a simple norm-based regularization is important and can help *adapt* to a dataset's intrinsic complexity yielding better generalization, thus partly explaining the empirical success of similar regularizations reported in previous works.

## 1 Introduction

In many machine learning tasks, data is represented in a high-dimensional Euclidean space. The $L_2$ distance in this space is then used to compare observations in methods such as clustering and nearest-neighbor classification. Often, this distance is not ideal for the task at hand. For example, the presence of uninformative or mutually correlated measurements arbitrarily inflates the distances between pairs of observations. *Metric learning* has emerged as a powerful technique to learn a *metric* in the representation space that emphasizes feature combinations that improve prediction while suppressing spurious measurements. This has been done by exploiting class labels [1, 2] or other forms of supervision [3] to find a Mahalanobis distance metric that respects these annotations.

Despite the popularity of metric learning methods, few works have studied how problem complexity scales with key attributes of the dataset. In particular, how do we expect generalization error to scale—both theoretically and practically—as one varies the number of informative and uninformative measurements, or changes the noise levels? In this work, we develop two general frameworks for PAC-style analysis of supervised metric learning. The *distance-based* metric learning framework uses class label information to derive distance constraints. The objective is to learn a metric that yields smaller distances between examples from the same class than those from different classes. Algorithms that optimize such distance-based objectives include Mahalanobis Metric for Clustering (MMC) [4], Large Margin Nearest Neighbor (LMNN) [1] and Information Theoretic Metric Learning (ITML) [2]. Instead of using distance comparisons as a proxy, however, one can also optimize for a specific prediction task directly. The second framework, the *classifier-based* metric learning framework, explicitly incorporates the hypotheses associated with the prediction task to learn effective distance metrics. Examples in this regime include [5] and [6].

Our analysis shows that in both frameworks, the sample complexity scales with a dataset's representation dimension (Theorems 1 and 3), and this dependence is necessary in the absence of assumptions about the underlying data distribution (Theorems 2 and 4). By considering any Lipschitz loss, our results improve upon previous sample complexity results (see Section 6) and, for the first time, provide matching lower bounds.

In light of our observation that data measurements often include uninformative or weakly informative features, we expect a metric that yields good generalization performance to de-emphasize such features and accentuate the relevant ones. We thus formalize the *metric learning complexity* of a given dataset in terms of the intrinsic complexity $d$ of the optimal metric. For Mahalanobis metrics, we characterize intrinsic complexity by the *norm* of the matrix representation of the metric. We refine our sample complexity results and show a *dataset-dependent* bound for both frameworks that relaxes the dependence on representation dimension and instead scales with the dataset's intrinsic metric learning complexity $d$ (Theorem 7).

Based on our dataset-dependent result, we propose a simple variation on the empirical risk minimizing (ERM) algorithm that returns a metric (of complexity $d$) that jointly minimizes the observed sample bias and the expected intra-class variance for metrics of fixed complexity $d$. This bias-variance balancing criterion can be viewed as a structural risk minimizing algorithm that provides better generalization performance than an ERM algorithm and justifies norm-regularization of weighting metrics in the optimization criteria for metric learning, partly explaining empirical success of similar objectives [7, 8]. We experimentally validate how the basic principle of norm-regularization can help enhance the prediction quality even for existing metric learning algorithms on benchmark datasets (Section 5). Our experiments highlight that norm-regularization indeed helps learn weighting metrics that better adapt to the signal in data in high-noise regimes.

## 2   Preliminaries

In this section, we define our notation, and explicitly define the distance-based and classifier-based learning frameworks. Given a $D$-dimensional representation space $X = \mathbb{R}^D$, we want to learn a weighting, or a *metric*[1] $M^*$ on $X$ that minimizes some notion of *error* on data drawn from a fixed unknown distribution $\mathcal{D}$ on $X \times \{0, 1\}$:
$$M^* := \operatorname{argmin}_{M \in \mathcal{M}} \operatorname{err}(M, \mathcal{D}),$$
where $\mathcal{M}$ is the class of weighting metrics $\mathcal{M} := \{M \mid M \in \mathbb{R}^{D \times D}, \sigma_{\max}(M) = 1\}$ (we constrain the maximum singular value $\sigma_{\max}$ to remove arbitrary scalings). For supervised metric learning, this *error* is typically label-based and can be defined in two intuitive ways.

The **distance-based framework** prefers metrics $M$ that bring data from the same class closer together than those from opposite classes. The corresponding distance-based error then measures how the distances amongst data violate class labels:
$$\operatorname{err}_{\mathrm{dist}}^\lambda(M, \mathcal{D}) := \mathbb{E}_{(x_1, y_1),(x_2, y_2) \sim \mathcal{D}}\Big[\phi^\lambda\big(\,\rho_{\mathrm{M}}(x_1, x_2), Y\big)\Big],$$
where $\phi^\lambda(\rho_{\mathrm{M}}, Y)$ is a generic distance-based loss function that computes the degree of violation between weighted distance $\rho_{\mathrm{M}}(x_1, x_2) := \|M(x_1 - x_2)\|^2$ and the label agreement $Y := \mathbf{1}[y_1 = y_2]$ and penalizes it by factor $\lambda$. For example, $\phi$ could penalize intra-class distances that are more than some upper limit $U$ and inter-class distances that are less than some lower limit $L > U$:
$$\phi_{L,U}^\lambda(\rho_{\mathrm{M}}, Y) := \left\{ \begin{array}{ll} \min\{1, \lambda[\rho_{\mathrm{M}} - U]_+\} & \text{if } Y = 1 \\ \min\{1, \lambda[L - \rho_{\mathrm{M}}]_+\} & \text{otherwise} \end{array} \right., \tag{1}$$

where $[A]_+ := \max\{0, A\}$. MMC optimizes an efficiently computable variant of Eq. (1) by constraining the aggregate intra-class distances while maximizing the aggregate inter-class distances. ITML explicitly includes the upper and lower limits with an added regularization on the learned $M$ to be close to a pre-specified metric of interest $M_0$.

While we will discuss loss-functions $\phi$ that handle distances between *pairs* of observations, it is easy to extend to relative distances among *triplets*:

$$\phi^\lambda_{\text{triple}}\big(\rho_{\text{M}}(x_1, x_2), \rho_{\text{M}}(x_1, x_3), (y_1, y_2, y_3)\big) := \begin{cases} \min\{1, \lambda[\rho_{\text{M}}(x_1, x_2) - \rho_{\text{M}}(x_1, x_3)]_+\} & \text{if } y_1 = y_2 \neq y_3 \\ 0 & \text{otherwise} \end{cases},$$

LMNN is a popular variant, in which instead of looking at all triplets, it focuses on triplets in local neighborhoods, improving the quality of local distance comparisons.

The **classifier-based framework** prefers metrics $M$ that directly improve the prediction quality for a downstream task. Let $\mathcal{H}$ represent a real-valued hypothesis class associated with the prediction task of interest (each $h \in \mathcal{H} : X \to [0, 1]$), then the corresponding classifier-based error becomes:

$$\text{err}_{\text{hypoth}}(M, \mathcal{D}) := \inf_{h \in \mathcal{H}} \mathbb{E}_{(x,y) \sim \mathcal{D}}\Big[ \mathbf{1}\big[|h(Mx) - y| \geq 1/2\big]\Big].$$

Example classifier-based methods include [5], which minimizes ranking errors for information retrieval and [6], which incorporates network topology constraints for predicting network connectivity structure.

## 3   Metric Learning Sample Complexity: General Case

In any practical setting, we estimate the ideal weighting metric $M^*$ by minimizing the empirical version of the error criterion from a finite size sample from $\mathcal{D}$. Let $S_m$ denote a sample of size $m$, and $\text{err}(M, S_m)$ denote the corresponding empirical error. We can then define the empirical risk minimizing metric based on $m$ samples as $M^*_m := \text{argmin}_M \text{err}(M, S_m)$, and compare its generalization performance to that of the theoretically optimal $M^*$, that is,

$$\text{err}(M^*_m, \mathcal{D}) - \text{err}(M^*, \mathcal{D}). \tag{2}$$

**Distance-Based Error Analysis.**   Given an i.i.d. sequence of observations $z_1, z_2, \dots$ from $\mathcal{D}$, $z_i = (x_i, y_i)$, we can pair the observations together to form a *paired* sample[2] $S^{\text{pair}}_m = \{(z_1, z_2), \dots, (z_{2m-1}, z_{2m})\} = \{(z_{1,i}, z_{2,i})\}^m_{i=1}$ of size $m$, and define the sample-based distance error induced by a metric $M$ as

$$\text{err}^\lambda_{\text{dist}}(M, S^{\text{pair}}_m) := \frac{1}{m} \sum_{i=1}^{m} \phi^\lambda\big(\rho_{\text{M}}(x_{1,i}, x_{2,i}), \mathbf{1}[y_{1,i} = y_{2,i}]\big).$$

Then for any $B$-bounded-support distribution $\mathcal{D}$ (that is, each $(x, y) \sim \mathcal{D}$, $\|x\| \leq B$), we have the following.[3],[4]

**Theorem 1** *Let $\phi^\lambda$ be a distance-based loss function that is $\lambda$-Lipschitz in the first argument. Then with probability at least $1 - \delta$ over an i.i.d. draw of $2m$ samples from an unknown $B$-bounded-support distribution $\mathcal{D}$ paired as $S^{\text{pair}}_m$, we have*

$$\sup_{M \in \mathcal{M}} \big[\text{err}^\lambda_{\text{dist}}(M, \mathcal{D}) - \text{err}^\lambda_{\text{dist}}(M, S^{\text{pair}}_m)\big] \leq O\left(\lambda B^2 \sqrt{D \ln(1/\delta)/m}\right).$$

This implies a bound on our key quantity of interest, Eq. (2). To achieve estimation error rate $\epsilon$, $m = \Omega((\lambda B^2/\epsilon)^2 D \ln(1/\delta))$ samples are sufficient, showing that one never needs more than a number proportional to $D$ examples to achieve the desired level of accuracy with high probability.

Since many applications involve high-dimensional data, we next study if such a strong dependency on $D$ is necessary. It turns out that even for simple distance-based loss functions like $\phi_{L,U}^\lambda$ (c.f. Eq. 1), there are data distributions for which one cannot ensure good estimation error with fewer than linear in $D$ samples.

**Theorem 2** *Let $\mathcal{A}$ be any algorithm that, given an i.i.d. sample $S_m$ (of size $m$) from a fixed unknown bounded support distribution $\mathcal{D}$, returns a weighting metric from $\mathcal{M}$ that minimizes the empirical error with respect to distance-based loss function $\phi_{L,U}^\lambda$. There exist $\lambda \geq 0$, $0 \leq U < L$ (indep. of $D$), s.t. for all $0 < \epsilon, \delta < \frac{1}{64}$, there exists a bounded support distribution $\mathcal{D}$, such that if $m \leq \frac{D+1}{512\epsilon^2}$,*

$$\mathbf{P}_{S_m}\left[ \mathrm{err}_{\mathrm{dist}}^\lambda(\mathcal{A}(S_m), \mathcal{D}) - \mathrm{err}_{\mathrm{dist}}^\lambda(M^*, \mathcal{D}) > \epsilon \right] > \delta.$$

While this strong dependence on $D$ may seem discouraging, note that here we made no assumptions about the underlying structure of the data distribution. One may be able to achieve a more relaxed dependence on $D$ in settings in which individual features contain varying amounts of useful information. This is explored in Section 4.

**Classifier-Based Error Analysis.** In this setting, we consider an i.i.d. set of observations $z_1, z_2, \dots$ from $\mathcal{D}$ to obtain the unpaired sample $S_m = \{z_i\}_{i=1}^m$ of size $m$. To analyze the generalization-ability of weighting metrics optimized w.r.t. underlying real-valued hypothesis class $\mathcal{H}$, we must measure the classification complexity of $\mathcal{H}$. The scale-sensitive version of VC-dimension, the *fat-shattering dimension*, of a hypothesis class (denoted $\mathsf{Fat}_\gamma(\mathcal{H})$) encodes the right notion of classification complexity and provides a way to relate generalization error to the empirical error at a *margin $\gamma$* [9].

In the context of metric learning with respect to a fixed hypothesis class, define the empirical error at a margin $\gamma$ as $\mathrm{err}_{\mathrm{hypoth}}^\gamma(M, S_m) := \inf_{h \in \mathcal{H}} \frac{1}{m} \sum_{(x_i, y_i) \in S_m} \mathbf{1}[\mathsf{Margin}(h(Mx_i), y_i) \leq \gamma]$, where $\mathsf{Margin}(\hat{y}, y) := (2y-1)(\hat{y} - 1/2)$.

**Theorem 3** *Let $\mathcal{H}$ be a $\lambda$-Lipschitz base hypothesis class. Pick any $0 < \gamma \leq 1/2$, and let $m \geq \mathsf{Fat}_{\gamma/16}(\mathcal{H}) \geq 1$. Then with probability at least $1 - \delta$ over an i.i.d. draw of $m$ samples $S_m$ from an unknown $B$-bounded-support distribution $\mathcal{D}$ ($\epsilon_0 := \min\{\gamma/2, 1/2\lambda B\}$)*

$$\sup_{M \in \mathcal{M}} \left[ \mathrm{err}_{\mathrm{hypoth}}(M, \mathcal{D}) - \mathrm{err}_{\mathrm{hypoth}}^\gamma(M, S_m) \right] \leq O\left( \sqrt{\frac{1}{m} \ln \frac{1}{\delta} + \frac{D^2}{m} \ln \frac{D}{\epsilon_0} + \frac{\mathsf{Fat}_{\gamma/16}(\mathcal{H})}{m} \ln \left( \frac{m}{\gamma} \right)} \right).$$

As before, this implies a bound on Eq. (2). To achieve estimation error rate $\epsilon$, $m = \Omega((D^2 \ln(\lambda DB/\gamma) + \mathsf{Fat}_{\gamma/16}(\mathcal{H}) \ln(1/\delta\gamma))/\epsilon^2)$ samples suffices. Note that the task of finding an optimal metric only additively increases sample complexity over that of finding the optimal hypothesis from the underlying hypothesis class. In contrast to the distance-based framework (Theorem 1), here we get a quadratic dependence on $D$. The following shows that a strong dependence on $D$ is necessary in the absence of assumptions on the data distribution and base hypothesis class.

**Theorem 4** *Pick any $0 < \gamma < 1/8$. Let $\mathcal{H}$ be a base hypothesis class of $\lambda$-Lipschitz functions that is closed under addition of constants (i.e., $h \in \mathcal{H} \implies h' \in \mathcal{H}$, where $h' : x \mapsto h(x) + c$, for all $c$) s.t. each $h \in \mathcal{H}$ maps into the interval $[1/2 - 4\gamma, 1/2 + 4\gamma]$ after applying an appropriate theshold.*

*Then for any metric learning algorithm $\mathcal{A}$, and for any $B \geq 1$, there exists $\lambda \geq 0$, for all $0 < \epsilon, \delta < 1/64$, there exists a $B$-bounded-support distribution $\mathcal{D}$ s.t. if $m \ln^2 m < O\left( \frac{D^2 + d}{\epsilon^2 \ln(1/\gamma^2)} \right)$*

$$\mathbf{P}_{S_m \sim \mathcal{D}}[\mathrm{err}_{\mathrm{hypoth}}(M^*, \mathcal{D}) > \mathrm{err}_{\mathrm{hypoth}}^\gamma(\mathcal{A}(S_m), \mathcal{D}) + \epsilon] > \delta,$$

*where $d := \mathsf{Fat}_{768\gamma}(\mathcal{H})$ is the fat-shattering dimension of $\mathcal{H}$ at margin $768\gamma$.*

# 4 Sample Complexity for Data with Un- and Weakly Informative Features

We introduce the concept of the *metric learning complexity* of a given dataset. Our key observation is that a metric that yields good generalization performance should emphasize relevant features while suppressing the contribution of spurious features. Thus, a good metric reflects the quality of individual feature measurements of data and their relative value for the learning task. We can leverage this and define the metric learning complexity of a given dataset as the *intrinsic complexity* $d$ of the weighting metric that yields the best generalization performance for that dataset (if multiple metrics yield best performance, we select the one with minimum $d$). A natural way to characterize the intrinsic complexity of a weighting metric $M$ is via the norm of the matrix $M$. Using metric learning complexity as our gauge for feature-set richness, we now refine our analysis in both canonical frameworks. We will first analyze sample complexity for norm-bounded metrics, then show how to *automatically adapt* to the intrinsic complexity of the unknown underlying data distribution.

## 4.1 Distance-Based Refinement

We start with the following refinement of the distance-based metric learning sample complexity for a class of Frobenius norm-bounded weighting metrics.

**Lemma 5** *Let $\mathcal{M}$ be any class of weighting metrics on the feature space $X = \mathbb{R}^D$, and define $d := \sup_{M \in \mathcal{M}} \|M^\mathsf{T} M\|_F^2$. Let $\phi^\lambda$ be any distance-based loss function that is $\lambda$-Lipschitz in the first argument. Then with probability at least $1 - \delta$ over an i.i.d. draw of $2m$ samples from an unknown $B$-bounded-support distribution $\mathcal{D}$ paired as $S_m^{\text{pair}}$, we have*

$$\sup_{M \in \mathcal{M}} \left[ \text{err}_{\text{dist}}^\lambda(M, \mathcal{D}) - \text{err}_{\text{dist}}^\lambda(M, S_m^{\text{pair}}) \right] \leq O\left( \lambda B^2 \sqrt{d \ln(1/\delta)/m} \right).$$

Observe that if our dataset has a low metric learning complexity $d \ll D$, then considering an appropriate class of norm-bounded weighting metrics $\mathcal{M}$ can help sharpen the sample complexity result, yielding a *dataset-dependent* bound. Of course, a priori we do not know which class of metrics is appropriate; We discuss how to *automatically adapt* to the right complexity class in Section 4.3.

## 4.2 Classifier-Based Refinement

Effective data-dependent analysis of classifier-based metric learning requires accounting for potentially complex interactions between an arbitrary base hypothesis class and the distortion induced by a weighting metric to the unknown underlying data distribution. To make the analysis tractable while still keeping our base hypothesis class $\mathcal{H}$ general, we assume that $\mathcal{H}$ is a class of two-layer feed-forward networks.[5] Recall that for any smooth target function $f^*$, a two-layer feed-forward neural network (with appropriate number of hidden units and connection weights) can approximate $f^*$ arbitrarily well [10], so this class is flexible enough to include most reasonable target hypotheses.

More formally, define the base hypothesis class of two-layer feed-forward neural network with $K$ hidden units as $\mathcal{H}_{\sigma^\gamma}^{\text{2-net}} := \{x \mapsto \sum_{i=1}^K w_i \, \sigma^\gamma(v_i \, \cdot \, x) \mid \|w\|_1 \leq 1, \|v_i\|_1 \leq 1\}$, where $\sigma^\gamma : \mathbb{R} \to [-1, 1]$ is a smooth, strictly monotonic, $\gamma$-Lipschitz activation function with $\sigma^\gamma(0) = 0$. Then, for generalization error w.r.t. any classifier-based $\lambda$-Lipschitz loss function $\phi^\lambda$,

$$\text{err}_{\text{hypoth}}^\lambda(M, D) := \inf_{h \in \mathcal{H}_{\sigma^\gamma}^{\text{2-net}}} \mathbb{E}_{(x,y) \sim \mathcal{D}} \left[ \phi^\lambda \big( h(Mx), y \big) \right],$$

we have the following.[6]

**Lemma 6** *Let $\mathcal{M}$ be any class of weighting metrics on the feature space $X = \mathbb{R}^D$, and define $d := \sup_{M \in \mathcal{M}} \|M^\mathsf{T} M\|_F^2$. For any $\gamma > 0$, let $\mathcal{H}_{\sigma\gamma}^{\text{2-net}}$ be a two layer feed-forward neural network base hypothesis class (as defined above) and $\phi^\lambda$ be a classifier-based loss function that $\lambda$-Lipschitz in its first argument. Then with probability at least $1 - \delta$ over an i.i.d. draw of $m$ samples $S_m$ from an unknown $B$-bounded support distribution $\mathcal{D}$, we have*

$$\sup_{M \in \mathcal{M}} \left[ \mathrm{err}_{\mathrm{hypoth}}^\lambda(M, \mathcal{D}) - \mathrm{err}_{\mathrm{hypoth}}^\lambda(M, S_m) \right] \leq O \left( B \lambda \gamma \sqrt{d \ln(D/\delta)/m} \right).$$

## 4.3 Automatically Adapting to Intrinsic Complexity

While Lemmas 5 and 6 provide a sample complexity bound tuned to the metric learning complexity of a given dataset, these results are *not* directly useful since one cannot select the correct norm-bounded class $\mathcal{M}$ a priori, as the underlying distribution $\mathcal{D}$ is unknown. Fortunately, by considering an appropriate sequence of norm-bounded classes of weighting metrics, we can provide a uniform bound that *automatically adapts* to the intrinsic complexity of the unknown underlying data distribution $\mathcal{D}$.

**Theorem 7** *Define $\mathcal{M}^d := \{M \mid \|M^\mathsf{T} M\|_F^2 \leq d\}$, and consider the nested sequence of weighting metric classes $\mathcal{M}^1 \subset \mathcal{M}^2 \subset \cdots$. Let $\mu_d$ be any non-negative measure across the sequence $\mathcal{M}^d$ such that $\sum_d \mu_d = 1$ (for $d = 1, 2, \cdots$). Then for any $\lambda \geq 0$, with probability at least $1 - \delta$ over an i.i.d. draw of sample $S_m$ from an unknown $B$-bounded-support distribution $\mathcal{D}$, for all $d = 1, 2, \cdots$, and all $M^d \in \mathcal{M}^d$,*

$$\left[ \mathrm{err}^\lambda(M^d, \mathcal{D}) - \mathrm{err}^\lambda(M^d, S_m) \right] \leq O \left( C \cdot B \lambda \sqrt{d \ln(1/\delta\mu_d)/m} \right), \tag{3}$$

*where $C := B$ for distance-based error, or $C := \gamma \sqrt{\ln D}$ for classifier-based error (for $\mathcal{H}_{\sigma\gamma}^{\text{2-net}}$).*

*In particular, for a data distribution $\mathcal{D}$ that has metric learning complexity at most $d \in \mathbb{N}$, if there are $m \geq \Omega \left( d(CB\lambda)^2 \ln(1/\delta\mu_d)/\epsilon^2 \right)$ samples, then with probability at least $1 - \delta$*

$$\left[ \mathrm{err}^\lambda(M_m^{\mathrm{reg}}, \mathcal{D}) - \mathrm{err}^\lambda(M^*, \mathcal{D}) \right] \leq O(\epsilon),$$

*for $M_m^{\mathrm{reg}} := \underset{M \in \mathcal{M}}{\arg\min} \left[ \mathrm{err}^\lambda(M, S_m) + \Lambda_M d_M \right]$, $\Lambda_M := CB\lambda \sqrt{\ln(\delta\mu_{d_M})^{-1}/m}$, $d_M := \left\lceil \|M^\mathsf{T} M\|_F^2 \right\rceil$.*

The measure $\mu_d$ above encodes our prior belief on the complexity class $\mathcal{M}^d$ from which a target metric is selected by a metric learning algorithm given the training sample $S_m$. In absence of any prior beliefs, $\mu_d$ can be set to $1/D$ (for $d = 1, \ldots, D$) for scale constrained weighting metrics ($\sigma_{\max} = 1$). Thus, for an unknown underlying data distribution $\mathcal{D}$ with metric learning complexity $d$, with number of samples just proportional to $d$, we can find a good weighting metric.

This result also highlights that the generalization error of *any* weighting metric returned by an algorithm is proportional to the (smallest) norm-bounded class to which it belongs (cf. Eq. 3). If two metrics $M_1$ and $M_2$ have similar empirical errors on a given sample, but have different intrinsic complexities, then the expected risk of the two metrics can be considerably different. We expect the metric with lower intrinsic complexity to yield better generalization error. This partly explains the observed empirical success of norm-regularized optimization for metric learning [7, 8].

Using this as a guiding principle, we can design an improved optimization criteria for metric learning that jointly minimizes the sample error and a Frobenius norm regularization penalty. In particular,

$$\min_{M \in \mathcal{M}} \quad \mathrm{err}(M, S_m) \quad + \quad \Lambda \|M^\mathsf{T} M\|_F^2 \tag{4}$$

for any error criteria 'err' used in a downstream prediction task and a regularization parameter $\Lambda$. Similar optimizations have been studied before [7, 8], here we explore the practical efficacy of this augmented optimization on existing metric learning algorithms in high noise regimes where a dataset's intrinsic dimension is much smaller than its representation dimension.

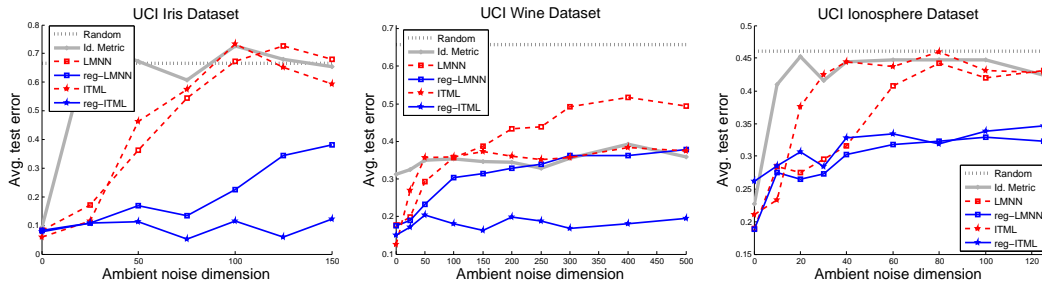

Figure 1: Nearest-neighbor classification performance of LMNN and ITML metric learning algorithms without regularization (dashed red lines) and with regularization (solid blue lines) on benchmark UCI datasets. The horizontal dotted line is the classification error of random label assignment drawn according to the class proportions, and solid gray line shows classification error of $k$-NN performance with respect to identity metric (no metric learning) for baseline reference.

## 5 Empirical Evaluation

Our analysis shows that the generalization error of metric learning can scale with the representation dimension, and regularization can help mitigate this by adapting to the intrinsic *metric learning complexity* of the given dataset. We want to explore to what degree these effects manifest in practice.

We select two popular metric learning algorithms, LMNN [1] and ITML [2], that are used to find metrics that improve nearest-neighbor classification quality. These algorithms have varying degrees of regularization built into their optimization criteria: LMNN implicitly regularizes the metric via its "large margin" criterion, while ITML allows for explicit regularization by letting the practitioners specify a "prior" weighting metric. We modified the LMNN optimization criteria as per Eq. (4) to also allow for an explicit norm-regularization controlled by the trade-off parameter $\Lambda$.

We can evaluate how the unregularized criteria (*i.e.,* unmodified LMNN, or ITML with the prior set to the identity matrix) compares to the regularized criteria (*i.e.,* modified LMNN with best $\Lambda$, or ITML with the prior set to a low-rank matrix).

**Datasets.** We use the UCI benchmark datasets for our experiments: IRIS (4 dim., 150 samples), WINE (13 dim., 178 samples) and IONOSPHERE (34 dim., 351 samples) datasets [11]. Each dataset has a fixed (unknown, but low) intrinsic dimension; we can vary the representation dimension by augmenting each dataset with synthetic correlated noise of varying dimensions, simulating regimes where datasets contain large numbers of uninformative features. Each UCI dataset is augmented with synthetic $D$-dimensional correlated noise as detailed in Appendix B.

**Experimental setup.** Each noise-augmented dataset was randomly split between 70% training, 10% validation, and 20% test samples. We used the default settings for each algorithm. For regularized LMNN, we picked the best performing trade-off parameter $\Lambda$ from $\{0, 0.1, 0.2, ..., 1\}$ on the validation set. For regularized ITML, we seeded with the rank-one discriminating metric, *i.e.,* we set the prior as the matrix with all zeros, except the diagonal entry corresponding to the most discriminating coordinate set to one. All the reported results were averaged over 20 runs.

**Results.** Figure 1 shows the nearest-neighbor performance (with $k = 3$) of LMNN and ITML on noise-augmented UCI datasets. Notice that the unregularized versions of both algorithms (dashed red lines) scale poorly when noisy features are introduced. As the number of uninformative features grows, the performance of both algorithms quickly degrades to that of classification performance in the original unweighted space with no metric learning (solid gray line), showing poor adaptability to the signal in the data.

The regularized versions of both algorithms (solid blue lines) significantly improve the classification performance. Remarkably, regularized ITML shows almost no degradation in classification perfor-

mance, even in very high noise regimes, demonstrating a strong robustness to noise. These results underscore the value of regularization in metric learning, showing that regularization encourages adaptability to the intrinsic complexity and improved robustness to noise.

# 6   Discussion and Related Work

Previous theoretical work on metric learning has focused almost exclusively on analyzing upper-bounds on the sample complexity in the distance-based framework, without exploring any intrinsic properties of the input data. Our work improves these results and additionally analyzes the classifier-based framework. It is, to best of our knowledge, the first to provide lower bounds showing that the dependence on $D$ is necessary. Importantly, it is also the first to provide an analysis of sample rates based on a notion of intrinsic complexity of a dataset, which is particularly important in metric learning, where we expect the representation dimension to be much higher than intrinsic complexity.

[12] studied the norm-regularized convex losses for *stable* algorithms and showed an upper-bound sublinear in $\sqrt{D}$, which can be relaxed by applying techniques from [13]. We analyze the ERM criterion directly (thus no assumptions are made about the optimization algorithm), and provide a precise characterization of when the problem complexity is independent of $D$ (Lm. 5). Our lower-bound (Thm. 2) shows that the dependence on $D$ is necessary for ERM in the assumption-free case.

[14] and [15] analyzed the ERM criterion, and are most similar to our results providing an upper-bound for the distance-based framework. [14] shows a $O(m^{-1/2})$ rate for thresholds on bounded convex losses for distance-based metric learning without explicitly studying the dependence on $D$. Our upper-bound (Thm. 1) improves this result by considering arbitrary (possibly non-convex) distance-based Lipschitz losses and explicitly revealing the dependence on $D$. [15] provides an alternate ERM analysis of norm-regularized metrics and parallels our norm-bounded analysis in Lemma 5. While they focus on analyzing a specific optimization criterion (thresholds on the hinge loss with norm-regularization), our result holds for general Lipschitz losses. Our Theorem 7 extends it further by explicitly showing when we can expect good generalization performance from a given dataset.

[16] provides an interesting analysis for *robust* algorithms by relying upon the existence of a partition of the input space where each cell has similar training and test losses. Their sample complexity bound scales with the partition size, which in general can be exponential in $D$.

It is worth emphasizing that none of these closely related works discuss the importance of or leverage the intrinsic structure in data for the metric learning problem. Our results in Section 4 formalize an intuitive notion of dataset's intrinsic complexity for metric learning, and show sample complexity rates that are finely tuned to this *metric learning complexity*. Our lower bounds indicate that exploiting the structure is necessary to get rates that don't scale with representation dimension $D$.

The classifier-based framework we discuss has parallels with the kernel learning and similarity learning literature. The typical focus in kernel learning is to analyze the generalization ability of linear separators in Hilbert spaces [17, 18]. Similarity learning on the other hand is concerned about finding a similarity function (that does not necessarily has a positive semidefinite structure) that can best assist in linear classification [19, 20]. Our work provides a complementary analysis for learning explicit linear transformations of the given representation space for arbitrary hypotheses classes.

Our theoretical analysis partly justifies the empirical success of norm-based regularization as well. Our empirical results show that such regularization not only helps in designing new metric learning algorithms [7, 8], but can even benefit existing metric learning algorithms in high-noise regimes.

## Acknowledgments

We would like to thank Aditya Menon for insightful discussions, and the anonymous reviewers for their detailed comments that helped improve the final version of this manuscript.

## Footnotes

[1]Note that we are looking at the linear form of the metric $M$; usually the corresponding quadratic form $M^\mathsf{T} M$ is discussed in the literature, which is necessarily positive semi-definite.

[2]While we pair $2m$ samples into $m$ independent pairs, it is common to consider all $O(m^2)$ possibly dependent pairs. By exploiting independence we provide a simpler analysis yielding $O(m^{-1/2})$ sample complexity rates, which is similar to the dependent case.

[3]We only present the results for paired comparisons; the results are easily extended to triplet comparisons.

[4]All the supporting proofs are provided in Appendix A.

[5]We only present the results for two-layer networks in Lemma 6; the results are easily extended to multilayer feed-forward networks.

[6]Since we know the functional form of the base hypothesis class $\mathcal{H}$ (*i.e.,* a two layer feed-forward neural net), we can provide a more precise bound than leaving it as $\text{Fat}(\mathcal{H})$.

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
