[Reviews · NeurIPS 2015]

Submitted by Assigned_Reviewer_1

I have one concern with the fact that in the proof of Theorem

2, lambda and L are set to depend on d. What are the

implications of this choice? Is it a realistic choice?

For a potentially useful additional citation, the following

paper considers the same loss as the one you have in (4), and

gives an efficient algorithm to globally solve it:

"Scalable Metric Learning for Co-embedding", Mirzazadeh et

al., 2015, ECML
Summary: This paper is clearly written and provides novel

theoretical insights into the sample complexity of metric

learning and improvements from added regularization. In

addition to carefully exploring the topic for both general and

specific cases theoretically, the empirical results are

similarly methodical but compact and further reinforce

these properties on real datasets.

Submitted by Assigned_Reviewer_2

This work proposes an optimization criteria that automatically adapts to the intrinsic complexity of the underlying dataset. Thus this method de-emphasize uninformative or weakly informative features and concentrates more on the relevant ones.

The paper is well written. The technical part is clear. It addresses an important issue in machine learning. This method seems novel to me although it might be even better to find the noise in the dataset and discard those items. Finally the experiments confirm their claim.
Summary: This work presents a novel method that adapts to the intrinsic complexity of dataset. The author show theoretically and empirically the better generalization of the algorithm.

Submitted by Assigned_Reviewer_3

This is the cleanest piece of work I have seen on sample complexity for metric learning and, as noted by the authors, presents several advantages over related work in this area.

In particular, I like that the authors present several important cases: both the "distance-based" and "classifier-based" frameworks, which highlight dependence on the dimensionality, as well as the regime in which there may be noisy or uninformative dimensions.

It's clearly written and appears to be correct.

Given that there is a relative dearth of work of this sort in the metric learning sub-community, I think it's worth publishing.

The experiments are somewhat underwhelming, though since this is more of a theoretical paper, it's not a key drawback.

A couple things I noticed were: there has actually been quite a bit of work on using the type of regularizer advocated in this work (the ||M^T M||_F^2 regularizer); see, e.g., "Metric Learning: A Survey", section 2.4.1 for examples).

It might be good to discuss these a bit.

Also, both ITML and LMNN can be viewed as already having a regularizer on the M matrix; for instance, ITML is typically viewed as applying a LogDet regularization on M^T M and LMNN is often viewed as applying a weighted trace norm regularization on M^T M.

Perhaps these were not necessarily the best choices of methods to add the proposed regularization?

Also, it's stated that ITML is initialized with a rank-one metric; given that the LogDet regularizer maintains rank (the rank cannot change from the initialization), this would lead to a low-rank solution, which would probably be undesirable (if I'm understanding the experimental setup correctly).

These are all fairly minor things, however, and shouldn't detract from the fact that the paper is a clear and effective look at sample complexity for metric learning.
Summary: This is a nice paper that presents several key results regarding the sample complexity for Mahalanobis metric learning.

It is a good addition to the literature on metric learning, and deserves to be published.

Submitted by Assigned_Reviewer_4

Summary In this article the authors present a theoretical analysis of non-regularized supervised metric learning formulations. They study two frameworks. On the one hand, the distance-based framework considers algorithms that optimize a distance w.r.t. class label information. On the other hand, the classifier-based framework considers algorithms which optimize a metric for a specific prediction or a ranking task.

For both frameworks, the authors provide PAC style bounds (sample complexity bounds) on the difference between the true risk and the empirical risk of the learned metric. They also analyze the difference between the true risk of the learned metric and the true risk of the best in-class metric and discuss the the necessity of a strong dependency on the dimensionality of the data. In the last part of the paper, the authors consider some regularized formulation using the Frobenius norm as a

regularizer, they propose some refinement of their results in this context and argue that adding a regularization term provides better guarantees. The authors propose finally an empirical evaluation of two well known metric learning approaches (namely ITML and LMNN) showing that adding a regularization term is indeed beneficial.

The paper is relatively clear and well written. The results are new and provide new theoretical understanding for metric learning .

Comments.

-Correct the proof of Lemma 2 to make it independent from the dimension. You should add some comments to make a link between Th1 and Th2. In the current form of the paper, they are presented separately and with formulations that do not allow the reader to directly get the consequences on the dependence on D.

-Mention that the results of Jin et al. (NIPS 2009) can be extended to replace the dependency on d by a term based on the data concentration (term B in author's paper), for any distributions and convex regularized formulations.

This can be done by using the techniques of Bousquet & Elisseeff, JMLR 2002 - note that the convexity of the formulation is mandatory for applying algorithmic stability. +Bellet et al. Metric Learning. Morgan & Claypool publishers. 2015. (chapter 8)

-One of the claim of the paper is to generalize some previous results to possibly non convex loss. However, in this context there is no guarantee to find the best model (matrix) minimizing the risk which makes the results a bit non informative on what can be done in practise.

Do not forget that even for convex formulation, taking into all the quadratic constraints involved by classical metric learning formulations is heavy and in general some heuristics are used to reduce the number of constraints.

One interesting perspective would be to know if the dependency on D impacts more severely non convex non regularized formulations.

-The authors have focused on the dependence over the representation dimension D, however it could be interesting to consider also the true rank of the matrix M rather than the data dimension. This would allow to make some relationships with metric learning based on Cholesky decomposition of the learned metric and methods making use of some low rank regularization. This point should be at least discussed in the perspectives.

-Considering, the classifier-based approach, the bibliography lacks a reference on methods learning a linear classifier from good similarity functions, even if this paper adresses other settings. +Balcan et al. Improved Guarantees for Learning via Similarity Functions. COLT 2008. +Bellet et al. Similarity Learning for Provably Accurate Sparse Linear Classification. ICML 2012. This aspect is also studied in the paper of Guo and Ying cited by the authors.

Overall, the results are interesting and I think that providing theoretical guarantees on classifiers making use of the result of a known metric is very important and results in that direction must continued.

The results are based on covering and fat shattering dimensions arguments, is there any perspective to use other frameworks (Rademacher-based for example?)

-About the experiments.

Even though they do not consider noisy data in their analysis, the authors create noise augmented datasets to see the influcence of the regularization. The conclusion of the experiments is that regularization can help to deal with noisy datasets or in other words they point out that the well known problem of overfitting can be partly adressed by choosing a good regularizer which is known fact. The scaling for the x-axis is different for the 3 figures, what is the reason for that? In particular, one can wonder what happens after 120/150 for Iris and Ionosphere? For the comparison with ITML, it could be interesting to have the accuracy obtained by directly using the rank-1 matrix alone in the distance to compare the influence of the regularizer. It seems that the choice of this matrix may do a lot of the "job". The experiments lack of datasets - 3 is very small and the number of dimensions is relatively low - and lack of methods for comparison. We can also wonder why no experiment on any classifier-based setting has been made. There also exist other datasets with higher dimensions in the UCI datasets, you can also check here : +Y. Shi et al.:Sparse Compositional Metric Learning. AAAI 2014: 2078-2084. The experiments would have gained in interest, if they authors have presented the evolution of the bounds with the data and with the dimension.

Other comments:

-line 075: the authors consider binary classification, however the loss considered below allows to consider a multiclass setting and some data sets in the experiments are multiclass. -I am wondering why in the definition of the true risk of the classifier-based approach, the margin is chosen at 1/2, while for the empirical risk the margin can be fixed to any value between 0 and 1/2. -line 219: "do now know" -> "do not know" -in the proof of Lemma 4, the notion of covering is used extensively.

Is it possible to extend the results to other regularization norms?
Summary: Pro:

-new and original bounds for Metric Learning

-analysis wrt to a notion of hypothesis space interesting

-analysis provided in the classifier-based setting Cons:

-limited experimental evaluation

-some discussions about regularization and links with other references lack of precision

Author Feedback
Author rebuttal: We thank the reviewers for their insightful comments.

R3 raises concern about the 'global conclusion' of this work, stating that 'regularized formulations have better generalization...and better practical behavior...is already known'

Similar to the observation by R3, we also discuss that norm-regularization is known to yield improved results in Metric Learning (ML) (lines 279-282). Our primary goal in this paper is instead to understand *why* this might be the case. We take a data-centric view, and characterize a dataset's intrinsic ML complexity in terms of the 'information content' in its features. We show that our proposed notion of intrinsic complexity helps in deriving sharper sample complexity rates for ML. To the best of our knowledge, no similar characterization of intrinsic complexity has been made before. Our result in Thm 7 shows an interesting connection: norm regularization is a simple way to *automatically adapt* to a dataset's intrinsic complexity (this relationship hasn't been rigorously studied before), thus providing a justification of why such types of regularizations are successful. Our work makes the connection between norm regularization and a dataset's 'information content' precise.
Along with making this connection, our paper:
- organizes the seemingly disparate complexity results in two important ML frameworks: "distance-based", and often overlooked, "classifier-based" framework.
- improves upon several previously derived results in a simple and clean way.
- provides lower bounds for the sample complexity in both the frameworks.
- provides both data independent and data dependent sample complexity results.

R3 and R4 raise concern about Thm 2 (lower bound) stating that the parameters L and lambda depend on the ambient dimension 'D', weakening the statement.

We thank the reviewers for this keen observation. This dependence was introduced accidentally while simplifying the proof. The proof can be altered easily to remove this dependence by considering a scaled version of the construction being used in the proof. Based on the way we do the reduction, the scale of the dataset does not change the sample complexity; and the proof still goes through. The theorem statement should be read as: "...There exist lambda > 0, 0 < = U < L (independent of D), such that for all...". This simple fix will be included in the final version of the manuscript. It is worth noting while we show this result for one setting of lambda,U & L, it is possible to show this for a large range of these parameters.
We would like to emphasize that this is the first time, to the best of our knowledge, any attempt has been made to provide lower bounds for ML sample complexity. Moreover, we believe that the proof of Thm 2 is not routine, and that the proof technique will be of independent technical interest.

R2 and R3 point out that the experiment section is 'weak'

As acknowledged by the reviewers, our focus in this paper is to derive PAC-style sample complexity results. Our goal with the experiments was to see to what extent the basic principle of norm regularization helps ML for datasets with fixed intrinsic complexity and changing representation dimension. As R3 and we (line 281) mention, regularizing the metric is known to help get better accuracies. What is not empirically well-studied is how robust these results are for changing representation dimension. Our experiments shed light on this: even when we deviate from the exact theoretical criterion --change the ERM to LMNN or ITML cost-- the regularization still helps automatically adapt to the intrinsic complexity.

Clarifications on minor comments:

- [R3, Thm 2 should be an "if and only if" result] Thm 2 (abbreviated) states "if m < D/eps^2, error > eps", and Thm 1 (abbreviated) implies "if m > D/eps^2, error < eps". The combination of two statements implies "if and only if".

- [R3, Jin et al. result can be extended to remove the dependency on D] As we mention in line 351, the original paper does not characterize the specific instances where this may be possible. Even after extending their results by applying techniques from Bousquet and Elisseeff (JMLR, 2002), the result only holds for convex regularized losses. Our result in Lemma 5 is for general (possibly non-convex) Lipschitz losses.

- [R3, optimization on non-convex losses is not practical] Our goal is to characterize the statistical sample complexity of the Metric Learning problem. Developing efficient algorithms is an important issue but not the focus of this paper.

- [R3, x-axis is different in the 3 figures] The prediction accuracy of vanilla LMNN and ITML for Iris and Ionosphere quickly saturated to that of random classification. There was no added gain in running the experiment for even higher noise dimensions.

- [R5, d appears on rhs but sup'ed over lhs in Thm 7] Thanks for pointing out this typo. Instead of 'sup', it should say "for all d and all M^d" in Eq 3.